# **Observational evidences of the influences of tropospheric** subtropical and midlatitude stratospheric westerly jets on the equatorial stratospheric intraseasonal oscillations

4 5

6

7

8

1

2

3

| G. Karthick Kumar Reddy <sup>1</sup> , T. K. Ramkumar <sup>2*</sup> and S. Venkatramana Reddy <sup>1</sup> |
|------------------------------------------------------------------------------------------------------------|
| 1. Dept. of Physics, Sri Venkateswara University, Tirupati 517502, India                                   |
| 2. National Atmospheric Research Laboratory, DOS, Govt. of India, Gadanki 517112, India                    |
| *Corresponding author: <u>tkram@narl.gov.in</u> , <u>tkramkumar@rediffmail.com</u>                         |

9

#### **Abstract:** 10

11 Using six Global Positioning System (GPS) Radio Occultation (RO) satellites (SAC-C, 12 METOP-A and COSMIC/FORMOSAT-3, CNOFS, GRACE and TerraSAR-X) determined 13 height profiles (1-40 km) of atmospheric temperature over the Indian tropical station of Gadanki 14 and the European Center for Medium Range Weather Forecast (ECMWF) Interim Reanalyses 15 (ERA-Interim) zonal wind and temperature data for four years (2009-2012), the present work reports that the tropospheric Subtropical Westerly Jet (SWJ) and the Midlatitude Stratospheric 16 17 Westerly Jet (MStWJ) play important roles in controlling differently the vertical propagation of 18 tropical Intra Seasonal Oscillations (ISO) with different period bands from the troposphere up to the stratosphere during Northern winters. In the months of December-May (Northern winter to 19 summer, NWTS) of all these years, there is significant 10-20 day and 20-40 day oscillations in 20 21 the troposphere up to the height of 13 km and above this it reappears at all heights above 21 km. 22 The 40-80 day oscillation also shows similar characteristics except that it almost disappeared 23 during NWTS months of the year 2010-2011 in the stratosphere. The absence of these signals in 24 the intervening heights of  $\sim$ 17-20 km is explained on the basis that these two bands actually 25 propagate from the tropical to subtropical region near the tropopause and then reappears in the 26 tropical stratosphere after refracted by the subtropical westerly jet. The poleward and 27 equatorward propagation of these bands in the troposphere and stratosphere respectively are 28 found using the ERA-interim data. Further the two longer period bands of ISO show strong

| 29                                                       | quasi-biennial oscillation in the lower atmosphere with opposite phases (when one band shows                                                                                                                                                                                                                                                                                                                                                                                                                                                                                                              |
|----------------------------------------------------------|-----------------------------------------------------------------------------------------------------------------------------------------------------------------------------------------------------------------------------------------------------------------------------------------------------------------------------------------------------------------------------------------------------------------------------------------------------------------------------------------------------------------------------------------------------------------------------------------------------------|
| 30                                                       | maximum the other one shows minimum in a particular year) between these two bands. It is also                                                                                                                                                                                                                                                                                                                                                                                                                                                                                                             |
| 31                                                       | observed that the phase of the tropical stratospheric Quasi Biennial Oscillation (QBO) has                                                                                                                                                                                                                                                                                                                                                                                                                                                                                                                |
| 32                                                       | significant control on the strength of the Mid latitude stratospheric westerly jet (MStWJ) that in                                                                                                                                                                                                                                                                                                                                                                                                                                                                                                        |
| 33                                                       | turn controls the refraction of the tropical tropospheric longer (40-80 days, Longer period ISO;                                                                                                                                                                                                                                                                                                                                                                                                                                                                                                          |
| 34                                                       | LISO) but not the smaller periods of ISO (SISO) back to the tropical stratosphere. In accordance                                                                                                                                                                                                                                                                                                                                                                                                                                                                                                          |
| 35                                                       | with earlier theoretical modelling studies, the westerly phase of the lower stratospheric QBO                                                                                                                                                                                                                                                                                                                                                                                                                                                                                                             |
| 36                                                       | occurred during NWTS months of 2010-2011 over the Indian longitudinal sector causes severe                                                                                                                                                                                                                                                                                                                                                                                                                                                                                                                |
| 37                                                       | disruption of the MStWJ at 30 km height. This disruption caused the prevention of refraction                                                                                                                                                                                                                                                                                                                                                                                                                                                                                                              |
| 38                                                       | back again to the tropical stratosphere of significant tropospheric LISO that arrived from the                                                                                                                                                                                                                                                                                                                                                                                                                                                                                                            |
| 39                                                       | tropics through the tropopause. Further, in these four years, it is observed no direct vertical                                                                                                                                                                                                                                                                                                                                                                                                                                                                                                           |
| 40                                                       | propagation of tropical tropospheric ISO to the stratosphere. The interannual variations in the                                                                                                                                                                                                                                                                                                                                                                                                                                                                                                           |
| 41                                                       | tropical stratospheric LISO are related strongly to the phase of the equatorial lower stratospheric                                                                                                                                                                                                                                                                                                                                                                                                                                                                                                       |
| 42                                                       | QBO in zonal wind and the strength of the MStWJ.                                                                                                                                                                                                                                                                                                                                                                                                                                                                                                                                                          |
| 43                                                       |                                                                                                                                                                                                                                                                                                                                                                                                                                                                                                                                                                                                           |
| 44                                                       |                                                                                                                                                                                                                                                                                                                                                                                                                                                                                                                                                                                                           |
| 45<br>46                                                 | Keywords: ISO in temperature and zonal wind; GPS RO and ERA-interim data; tropical                                                                                                                                                                                                                                                                                                                                                                                                                                                                                                                        |
| 47                                                       | tropospheric and stratospheric temperature and winds; seasonal characteristics of MJO                                                                                                                                                                                                                                                                                                                                                                                                                                                                                                                     |
| 10                                                       |                                                                                                                                                                                                                                                                                                                                                                                                                                                                                                                                                                                                           |
| 40                                                       |                                                                                                                                                                                                                                                                                                                                                                                                                                                                                                                                                                                                           |
| 49                                                       | Introduction                                                                                                                                                                                                                                                                                                                                                                                                                                                                                                                                                                                              |
| 49<br>50                                                 | Introduction                                                                                                                                                                                                                                                                                                                                                                                                                                                                                                                                                                                              |
| 49<br>50<br>51<br>52                                     | Introduction                                                                                                                                                                                                                                                                                                                                                                                                                                                                                                                                                                                              |
| 49<br>50<br>51<br>52<br>53                               | Introduction<br>Using the wind velocities measured by a medium frequency radar over the Christmas                                                                                                                                                                                                                                                                                                                                                                                                                                                                                                         |
| 49<br>50<br>51<br>52<br>53<br>54                         | Introduction . Using the wind velocities measured by a medium frequency radar over the Christmas island, it has been reported the existence of Intra seasonal oscillations (ISOs) with peaks near the                                                                                                                                                                                                                                                                                                                                                                                                     |
| 49<br>50<br>51<br>52<br>53<br>54<br>55                   | Introduction . Using the wind velocities measured by a medium frequency radar over the Christmas island, it has been reported the existence of Intra seasonal oscillations (ISOs) with peaks near the periods of ~ 60 days, 35 - 40 days and 22 - 25 days in the equatorial Mesosphere and Lower                                                                                                                                                                                                                                                                                                          |
| 49<br>50<br>51<br>52<br>53<br>54<br>55<br>56             | Introduction<br>Using the wind velocities measured by a medium frequency radar over the Christmas<br>island, it has been reported the existence of Intra seasonal oscillations (ISOs) with peaks near the<br>periods of ~ 60 days, 35 - 40 days and 22 - 25 days in the equatorial Mesosphere and Lower<br>Thermosphere (MLT) region winds (Eckermann and Vincent, 1994; Eckermann et al., 1997). It                                                                                                                                                                                                      |
| 49<br>50<br>51<br>52<br>53<br>54<br>55<br>56<br>57       | Introduction<br>Using the wind velocities measured by a medium frequency radar over the Christmas<br>island, it has been reported the existence of Intra seasonal oscillations (ISOs) with peaks near the<br>periods of ~ 60 days, 35 - 40 days and 22 - 25 days in the equatorial Mesosphere and Lower<br>Thermosphere (MLT) region winds (Eckermann and Vincent, 1994; Eckermann et al., 1997). It<br>is also reported similar periodicities in gravity-wave and diurnal-tide amplitudes in the MLT                                                                                                     |
| 49<br>50<br>51<br>52<br>53<br>54<br>55<br>56<br>57<br>58 | Introduction<br>Using the wind velocities measured by a medium frequency radar over the Christmas<br>island, it has been reported the existence of Intra seasonal oscillations (ISOs) with peaks near the<br>periods of ~ 60 days, 35 - 40 days and 22 - 25 days in the equatorial Mesosphere and Lower<br>Thermosphere (MLT) region winds (Eckermann and Vincent, 1994; Eckermann et al., 1997). It<br>is also reported similar periodicities in gravity-wave and diurnal-tide amplitudes in the MLT<br>region (Eckermann et al., 1997). It was suggested that the 30 - 60 day Madden-Julian oscillation |

separate 20 - 25 day oscillation over the western pacific (Hartmann et al., 1992) can modulate the 60 61 gravity wave and diurnal-tidal intensities at ISO periods. These modulated waves and tides then 62 propagate upwards to the MLT region and create similar periodicities in the wave-induced zonal 63 mean flow of the MLT region. Using observations of the winds from HRDI (High Resolution 64 Doppler Imager), Lieberman (1998) showed that the ISO in the MLT zonal winds is wavedriven. Rao et al. (2009) considered radar observations of MLT region dynamics in different 65 66 longitudinal sectors and found that ISO amplitudes have significant longitude variation, 67 suggesting a close connection to the vigor of convective activity in the underlying troposphere. Earlier, it is reported that the upper mesospheric (based on an empirical analysis of 68 69 measurements with the High Resolution Doppler Imager (HRDI) on the Upper Atmospheric 70 Research Satellite (UARS) spacecraft) intra seasonal oscillations (2-4 months periodicity) in the 71 meridional winds are associated with similar oscillations occurring in temperature (Microwave 72 Limb Sounder (MLS) of UARS data) near the stratospheric height of ~55 km (Huang and Reber, 73 2003; Huang et al., 2005). Recently, Rokade et al. (2012) showed, using medium frequency radar 74 data in the Indian tropical stations of Tirunelveli and Kolhapur, the intraseasonal oscillation 75 (ISO) in the mesospheric winds are correlated well with ISO in the outgoing longwave radiation, indicating that the mesospheric ISOs are inherently related to the ISOs in the tropospheric 76 77 convective activities (say Madden Julian Oscillation).

78

It is also reported the biennial variations of ISO in the zonal winds and diurnal tide amplitude in the mesosphere (Isoda et al., 2004). Further, Isoda et al. (2004) reported that gravity wave variances at ISO periods and the ISO in the zonal wind are not correlated in the MLT region. They suggested that dissipating non-migrating diurnal tide modulated at the ISO period is

the source of ISO in the MLT region zonal wind. . Using UARS/HRDI zonal wind data Lieberman [1998] studied the characteristics of ISO in the MLT region during December 1991 -March 1995.. Peak ISO amplitude of about 20 m/s was found at 95 km and 75 km in the latitudes of  $\pm 20^{\circ}$  around the equator, with a local minimum at around 80 km. By using a GCM, Miyoshi and Fujiwara (2003) showed variations in the migrating diurnal tide amplitudes with periods of 12 and 25 days in the heights from 20 to 300 km, implyingdynamical coupling between the lower and upper atmospheres.

90

The important characteristic of the Madden Julian Oscillation (MJO) in the tropical 92 troposphere is that it attenuates rapidly above the tropopause (Madden and Julian, 1971). This 93 was corroborated by studies of rocket and radiosonde data from the Indian tropical stations 94 (Nagpal and Raghavarao, 1991; Nagpal et al., 1994; Kumar and Jain, 1994). Using rocket wind 95 data from Indian stations spanning 8.5° - 21.5°N, the reintensification of the MJO in the upper stratosphere was reported by Nagpal and Raghavarao (1991), Nagpal et al. (1994) and Kumar 96 97 and Jain (1994). The occurrence of nearly equal amounts of intra seasonal activity in both the 98 zonal and meridional winds led them to conclude that its characteristics are different from that of 99 Kelvin waves, and associated the activity with Rossby waves. Further it was argued that these 100 Rossby waves would have propagated into the tropical upper stratosphere through mid latitudes 101 (Nagpal and Raghavarao, 1991; Nagpal et al., 1994). However, Kumar and Jain (1994) suggested 102 that the waves would have leaked into the upper stratosphere from the tropical troposphere via 103 direct vertical propagation because the MJO contains a significant Rossby-wave component, 104 particularly in forcing zones near India (Hendon and Salby, 1994). Mote et al. (2000) showed 105 signatures of tropical intraseasonal oscillation in the upper tropospheric moisture and dynamical

fields in the heights of 100-200 hPa levels with complicated characteristics near the 100 hPa 107 level. For the first time on a global scale, their analysis demonstrates that the response of the 108 100-hPa level water vapor to the Tropical Intra seasonal oscillation (TIO) is out of phase with 109 that at 215 and 147 hPa levels. They also found that while the convectively active phase 110 moistens the upper troposphere, the tropopause region becomes dryer. Madden and Julian (1972) 111 observed significant MJO in temperature near 100 hPa level. Mote et al. (1998) noted a 30- to 112 60-day spectral peak in water vapor measured by Microwave Limb Sounder (MLS). The sharp attenuation of the TIO signal in the lower stratosphere was noted also by Mote et al. (2000). 113 114 Further it was observed that the spectral power of water vapor variations in the 30- to 70- day 115 band drops by an order of magnitude between 100 and 68 hPa (Mote et al., 1998).

It is interesting to note some other aspects of the ISO. It is found that ISO period of 60– 119 80-days is present not only in the atmospheric dynamical parameters but also in the fluxes of solar ultra violet (UV) rays and ozone concentration (Zhou et al., 1997). Moreover, the 120 121 intraseasonal oscillations are found not only in the equatorial stratosphere but also over the 122 Antarctica during Austral winters. Based on the Met Office stratospheric assimilated data and the 123 TOMS total ozone, Huang and Weng (2002) reported that Stratospheric Antarctic Intraseasonal 124 Oscillation (SAIO) (30-day oscillation) occurs (within 60E - 120E) with deep vertical structure extending from the upper troposphere to the upper stratosphere. They found that the amplitude 125 126 increases rapidly with height below 5 hPa and decreases slowly with height above and the 127 vertical shows westward-tilting with increasing height below 5 hPa and a more barotropic 128 structure above. The calculated vertical and meridional wavelengths are about 80 km and 13,343

129 km respectively, and the scale height is 7 km, mostly westward propagating. Further, they 130 suggested that the topographically forced planetary wave propagates upward in a baroclinic 131 atmosphere and only those waves with largest zonal scale can propagate deep into the upper 132 stratosphere because of the Charney-Drazin criterion (Charney and Drazin, 1961; Karoly and 133 Hoskins, 1982).

The main aim of the present study is that to show how different bands of ISO (10-20 days, 20-40 days and 40-80 days) propagate to the tropical stratosphere from below in the 136 137 troposphere through the subtropical jet over the Indian region. oscillation Analysing Microwave 138 Sounding Unit (MSU4) and Stratospheric Sounding Unit (SSU) temperature data sets, Ziemke 139 and Stanford (1990) found that there is no direct vertical propagation of 1-2 month signals in the equatorial stratosphere. Further, Ziemke and Stanford (1991), Niranjan Kumar et al. (2011) and 140 141 Guharay et al. (2014) have clearly shown that the tropical stratospheric ISO is a result of vertical 142 propagation of ISO from the lower troposphere to tropopause from where it gets refracted to mid 143 latitudes. The presence of the subtropical westerly jet allows the equatorial ISO to get refracted 144 both vertically and laterally towards again to the equatorial region. The focus of the present study is to show the strong and direct link between the interannual modulation of the tropical 145 146 stratospheric ISO and the interannual variation of the strength of different ISO bands in the 147 troposphere Using six GPSRO satellites determined vertical profiles of temperature in the 148 heights of 6-40 km over the Indian tropical station of Gadanki and the ERA-interim reanalyses 149 temperature as well as the zonal winds in the whole tropical-high latitude regions, the present 150 work attempts to establish the concept that strong subtropical westerly jet allows the easier 151 subsequent propagation of the refracted tropical tropospheric ISO back to the equatorial

stratosphere for the years 2009-2012. It is natural to get interested to know that when some 153 atmospheric physical phenomena (say intra seasonal oscillations) are normally explained by 154 available large scale data sets (NCEP-NCAR, ERA-interim, MERRA etc.) what actually 155 happens in a limited regional space that is a small subset of larger domain in which large scale 156 physics (ISO) is happening. After all large scale phenomena can be considered as ensemble 157 average of small scale phenomena occurring in small size regions. One can easily ask whether 158 these large scale phenomena (ISO) can be seen in smaller scale regions like the present case of 159 Gadanki. Further, question arises as to whether an ensemble average of many small-time-scale 160 phenomena (wave-mean flow interaction of gravity waves, tides, Kelvin waves, planetary waves 161 etc. whose periods are within a few to tens of days) or a single large-time-scale phenomena as a 162 whole contributes to the observation of intra seasonal oscillation. The best example here is the 163 persistent tropical stratospheric quasi-biennial oscillation occurring due mainly to wave-mean 164 flow interaction. The present work is the result of these basic questions. However, in order to 165 show that the ISO as a whole propagates poleward near the tropppause and equatorward at higher heights through refraction about the subtropical jet, the present work includes analyses of 166 167 large ERA-interim data sets of temperature and zonal wind that cover the full latitude zone of 0-30N and the full height of 1-40 km in these years of 2009-2012. 168

While section 2 describes briefly about the six GPS RO satellites determined height profiles of temperature over Gadanki and the ERA-interim reanalyses temperature and zonal wind data, section 3 gives the detailed observations of the characteristics of ISO in particularly three bands (1) 10-20 day oscillation, (2) 21-40 day oscillation and (3) 41-80 day oscillation in the whole heights of the tropical troposphere and stratosphere. Section 4 provides a detailed

discussion of the observations and the section 5 gives the summary and conclusion of the present

- work.

#### 178 Data and Methodology:

Global Positioning System (GPS) satellites based radio occultation technique has become 181 a powerful remote sensing tool to determine the vertical profiles of atmospheric refractivity, 182 geopotential, temperature, pressure, water vapor, etc. with high accuracy and vertical resolution 183 (0.5 km to 1 km) in the Upper Troposphere -Lower Stratosphere (UTLS) and ionospheric 184 electron density (Kursinski et al., 1997; Rocken et al., 1997; Steiner et al., 1999). Global Positioning System /Meteorology (GPS/Met) successfully demonstrated the applicability of the 185 186 RO technique for probing the Earth's atmosphere and the provision of accurate data [Ware et al., 187 1996; Rocken et al., 1997; Steiner et al., 1999]. Information about these GPSRO satellites, namely, SAC-C (for lower atmospheric temperature studies see for e.g. Schmidt et al., 2005), 188 METOP-A and COSMIC/FORMOSAT-3, CNOFS (for lower atmospheric temperature studies 189 190 see for e.g. Wang et al., 2015), GRACE and TerraSAR-X can be found in these web pages: https://eoportal.org/web/eoportal/satellite-missions/s/sac-c, 191 192 https://eoportal.org/web/eoportal/satellite-missions/m/metop, 193 https://eoportal.org/web/eoportal/satellite-missions/f/formosat-3, 194 https://eoportal.org/web/eoportal/satellite-missions/c-missions/cnofs, 195 https://eoportal.org/web/eoportal/satellite-missions/g/grace. https://eoportal.org/web/eoportal/satellite-missions/t/terra. 196 197 In the present work, we utilized the temperature (wet temperature, wetPrf) information obtained 198 from these six satellites in the whole height region of 6-40 km and for the latitudinal and longitudinal regions of 80-180N and 750-850E centered about the Indian tropical station of 199 National Atmospheric Research Laboratory (NARL), Gadanki (13.5°N, 79.2°E) during the years 200 2009-2012 to determine the vertical propagation characteristics of planetary scale waves and 201 202 oscillations (10-120 day oscillation). This product wetPrf is based on a 1DVAR retrieval using

203 dry RO temperatures and ECMWF data (for details see the description on the UCAR data 204 website). Daily height profiles of temperature are constructed for the full year of 2009-2012 over 205 the mentioned coordinates by averaging the temperature values obtained within this region by all 206 the satellites. Small data gaps are linearly interpolated in time (days). All the data presented in 207 the work and the detailed information about these satellites is obtained freely from the website of 208 http://cdaac-www.cosmic.ucar.edu/cdaac/products.html. Since the radio occultation (RO) 209 technique gives the most accurate (~0.05 K in the heights of 8-30 km) measurement of 210 atmospheric dry temperature with high vertical resolution (increasing from ~60 m near the 211 surface to ~1.5 km at 40 km), it is being taken as the bench mark measurement against all other 212 measurements made with radiosonde sensors and satellite based brightness temperature at 213 microwave or infrared frequencies. When we study the characteristics of long period oscillations 214 (say intraseasonal oscillation) in different years, it is essential that there should not be any errors 215 in the measurements as it happens with different types of sensors in radiosondes and different 216 weighting functions of height associated with different frequencies of brightness temperature in 217 the case of satellites. It is to be remembered that RO technique is mission and geography 218 independent (Kuo et al., 2004 and 2005; Ho et al. 2007; He at al., 2009; Ho et al., 2009). Since the sampling errors associated with the present considered geographical grid and the number of 219 220 measurements available at different times in a day within the grid is significantly less with 221 respect to long period intraseasonal oscillations, they are not shown or discussed here.

222

For the detailed information on model reanalysis data of ERA-interim reanalysis data one is referred to Dee *et al.* (2011) and the present data at different pressure levels are downloaded from the website <u>http://apps.ecmwf.int/datasets/data/interim\_full\_daily/?levtype=pl</u>. In the

| 226                                    | present work, Morlet wavelet transform mehod has been employed for the detailed investigation                   |
|----------------------------------------|-----------------------------------------------------------------------------------------------------------------|
| 227                                    | of time evolution characteristics of oscillations with period bands of particularly the 10-20, 21-40            |
| 228                                    | and 40-80 days.A practical step-by-step guide to wavelet analysis with examples time series is                  |
| 229                                    | provided by Torrence and Compo (1998), including statistical significance testing and cone of                   |
| 230                                    | influence (COI) for the continuous wavelet transform. The complete knowledge of the wavelet                     |
| 231                                    | analyses carried out in the present study is guided by Torrence and Compo (1998).                               |
| 232<br>233<br>234<br>235<b