# Peer review of "Observational evidences of the influences of tropospheric subtropical and midlatitude stratospheric westerly jets on the equatorial stratospheric intraseasonal oscillations"

_Atmospheric Chemistry and Physics, 2016_

## Referee Comment (RC1) · P. Haynes (Referee) · 28 May 2016

(I have identified myself as referee because this is an unusual case in that I am acting as a referee as well as Editor.)

This paper considers intraseasonal oscillations - defined as oscillations with periods 10-80 days, with further subdivision into 10-20, 20-40 and 40-80 day periods, primarily in the tropics, but including the connections to the extra tropics.

The scientific rationale can be summarised as follows. Intraseasonal oscillations are excited by low-frequency variability in the tropical troposphere, in particular the

Madden-Julian Oscillation. These oscillations are observed in the tropical stratosphere (and higher) but only above about 20km, i.e. they are absent from the lowest part of the stratosphere. The explanation is that the oscillations excited in the troposphere are refracted into mid-latitudes in the upper troposphere, so are absent in the tropical lower stratosphere, and are then refracted back into the tropics in the stratosphere, so reappear in the tropic at some level. This explanation of observations has previously been provided by Ziemke and Stanford (1991) (who focus on the southern hemisphere and consider levels up to 40km or so) and Niranjankumar et al (2011) (who focus on the northern hemisphere and consider a combination of radar, lidar and satellite data and reanalysis data, which extends up to 90km).

The new aspect of the paper under consideration is that it considers interannual variations in the oscillation observed in the tropical stratosphere and relates them to variations in the winds in the midlatitude stratosphere. The years considered are 2009-2012, and 2011 is identified as an anomalous year. The data used is radio occultation temperature data (but restricted to a NH tropical region over India) together with ERA-interim reanalysis data.

I find this paper unsatisfactory for the following reasons:

(i) The new ingredient (relative to previous publications) is interannual variability, but only 4 years are considered. Therefore any identification of behaviour is highly speculative. (There is no modelling to support any of the ideas presented.)

(ii) It is difficult to see the value of the radio occultation data in this study. There is no argument that the phenomenon being studied has small vertical scales and therefore can be studied much more effectively with radio occultation data than with reanalysis data. Furthermore the radio occultation data is used only in a limited geographical region. Why has it not been used across the whole tropics and subtropics? (It seems as though the region of study has been used for historical reasons, this is for example that has been observed by an MST radar, but there is no use of radar data here.)

(iii) Given the absence of any arguments that 'new' data such as radio occultation data is essential, there would apparently be no reason not to use ERA-interim data (for example) over the whole 30-year time period for which it is available - and that would be a much more satisfactory approach to studying inter annual variability. It would also allow various types of composite/correlation analysis of the type used by the two papers mentioned above.

(iv) The paper is not well written. I have given many comments below on this. I note in particular the large number of abbreviated terms defined. If the authors were to insist on retaining these then perhaps there should be a table summarising the definitions. Additionally there are a very large number of figures and these are ineffective in identifying and explaining the important points that the authors wish to make. (The number of the figures and their nature is simply unhelpful to the reader.)

Therefore my recommendation as a referee is that this paper should be rejected for publication in ACP.

Detailed comments:

l16: 'MStWJ' - is this intended to be distinct from the stratospheric polar night jet?

l19: 'December-May (Northern winter to summer, NWTS)' - this is one of many abbreviated terms that are introduced - my own view is that there are too many. In any case they need to be chosen to be as intuitive as possible - Northern winter and spring - NWS - might be better. But I suggest you simply say 'December-May' when needed.

l20: 'there is significant' > 'there are significant'

l22: 'The 40-80' > 'A 40-80' (because a 40-80 oscillation has not previously been introduced, nor, I believe, is it a standard term).

l26: 'refracted' > refraction'

l28: 'the two longer period bands' - be explicit - 'the 20-40 and 40-80 period bands'

l30-32: 'It is also observed that the phase of the . . . QBO . . . has significant control on the strength of the . . . MStWJ . . . that in turn controls the refraction' - I don't see how you can argue on the basis of the 4 years of observations presented (and nothing else) that the QBO has significant control on the stratospheric jet, nor do I see how you can argue that the stratospheric jet has control over refraction.

l33-34: 'LISO' and 'SISO' - two new abbreviations which in my opinion do not help the reader.

l54: 'island' > 'Island'

l54: 'Intra Seasonal Oscillations' - say explicitly what you mean by this term - I think any oscillations with period 10-80 days.

l53-77: There is a lot of detail here about the mesosphere, which is not the main subject of the paper. Unless you can identify specific aspects of the mesosphere which relate closely to the method or conclusions of the paper then this material should be significantly shortened - as it stands it distracts the reader from what turns out to be the main focus of the paper. The same comment applies to l79-89 - though some of that relates to a possible QBO effect and is therefore a little more relevant.

l64: 'It is reported' > 'It was reported'

l82: 'that dissipating' > 'that the dissipating'

l95-l97: You leap from 'MJO' in l95 to 'intraseasonal activity' in l97. Is that intended to make a distinction - e.g. is 'intraseasonal activity' broader then 'MJO'?

l108-110: 'response of 100-hPa level water vapour . . . is out of phase with that at 215 and 147hPa levels' - 'convectively active phase moistens the upper troposphere, the tropopause region becomes dryer' - aren't those two things the same?

l112-115: It wasn't clear to me how this information on the water vapour signal in the lower stratosphere was relevant to what you are considering (which is more to do with

the dynamics). All in all, this paragraph gives the sense of a slightly random set of facts about the MJO. More focus would help the reader.

l118-113: Again it wasn't clear to me that this material on high-latitude intraseasonal oscillations was relevant. Are you implying a physical connection/relevance to the low-latitude phenomenon - or is this simply describing another 'intraseasonal' (=low-frequency) oscillation?

l140: You refer here to the previous work of Ziemke and Stanford (1991) who considered an analogous problem in the southern hemisphere and Niranjan Kumar et al (2011) who considered vertical propagation of intraseasonal oscillations in the northern hemisphere. What I am missing is what new ingredients you are providing over the Niranjan Kumar et al (2011) paper.

l143: I think that by 'the subtropical westerly jet' here you mean a jet in the stratosphere - please be explicit.

l149: 'whole tropical-high latitude regions' - clearer to say 'in the whole Northern Hemisphere'?

l152-168: This paragraph seems to a justification of presenting the information local to Gadanki as part of a study of what is a much larger scale phenomenon. You mention the QBO as an analogous example, but the analogy seems weak - the QBO is a large-scale phenomenon that requires small-scale processes for its existence. I don't see any argument in your paper as written that the large-scale phenomenon of intraseasonal oscillations requires small-scale processes (that may be observed local to Gadanki) for its existence.

l197: 'wet temperature, wetPrf' - this is important technical detail but most readers will find it mystifying. Provide a reference that gives more information on these terms.

l199: GPS radio occultation potentially provides temperature observations over much larger regions than that you have chosen. Why did you not use all the available observations?

l265: Chen and Robinson (1992) considered the combined effects of vertical shear and change in buoyancy frequency at the extratropical tropopause. So I don't see that their paper can be used straightforwardly to explain what may or may not be happening at the tropical tropopause.

l268: You should mention the black lines in the Figures in the captions as well as the text. If one set of black lines is supposed to indicate April and the other December then you should differentiate between the two - e.g. by making one set dashed.

l292: 'near 120 days in 2009' - but this feature lies outside your bounding curves (at left=hand and right-hand ends of the figure) for the validity of the wavelet transform calculation. So how can you consider this feature to be important?

l295: 'this oscillation is present' - which oscillation exactly - ∼50 days or ∼120 days? You say 'at all other heights'. But at several heights, e.g. 38km and 40km, there are no crimson contours, indicating significance in all non-2011 years. (It would be very helpful to have years marked on the Figures and I do not understand why this has not been done.)

l302: 'distinctly enhanced 40-80 day band oscillation ... during the months of November-May in all the years' - only in 2 years surely. As you say yourself the signal is weak in 2011 (and, if 2009 is to be considered to be within the range of the data, there is very little signal in 2009).

l328: 'zonal wind shows clear poleward propagation (tilting towards the right side with increasing latitude)' - this tilt - which I suppose should be visible in the bottom-left panel of Figure 8 - seems very weak to me. You note that the wind fluctuations (apparently) show phase tilt, but the temperature does not. How do you rationalise that with a dynamical mechanism? Are the temperature and wind signals independent - i.e. showing essentially independent dynamical phenomena? Are both signals 'real'?

l343: You have previously described a phenomenon of refraction back towards the tropics above 19km (see ll317-321). But here you are using the wind at 30km as an explanation for absence of refraction in 2011. Why should the wind at 30km affect what is happening at 19km upwards?

l346: It is not clear to me that invoking the Arctic Oscillation (in discussing the wind variations at 10hPa) here is very helpful. The Arctic Oscillation is primarily a tropospheric phenomenon in which the mid-latitude eddy-driven jet shifts (with implications for, e.g. sea-level pressure at high latitudes). It is true that stratospheric variations are discussed in terms of an NAM (Northern Annular Mode), but that is simply the dominant pattern of dynamical variability, in the stratosphere or at some other chosen level. In the troposphere the NAM is associated with the Arctic Oscillation, but that does not imply that the association can be made at all levels.

l348: You seem to be considering the zonal velocity at 70hPa simply to show that it does not show anomalous behaviour in 2011 relative to other years, in contrast to the zonal velocity at 10hPa.

l354: 'no significant LISO in . . . 2011 in the heights of 30-40km' - if the intention is to focus on what is happening above 30km then that should have been made clear at the beginning of this paragraph. But the fact is that you have previously characterised the absence of LISO by referring to Fig 4b - in which inter annual differences are shown in the layer 19-32km.

l358: '50-day oscillation' - are you using this term to be intentionally different from previous use of '40-80-day'?

l383: You have focused here on the phase behaviour of the 40-80 day fluctuations. But one thing that is striking about Figs 12-15 is that the amplitude (e.g. in Jan-May) shows no sign of the claimed disappearance of the oscillation in the tropics in a region just above the tropopause. So how is the amplitude behaviour you show consistent with your previous discussion and indeed with the theme of the paper as a whole.

l413: 'increasing trend' is a very confusing term to use to describe increase with height.

l460: 'in the first few kilometers of height' - be more precise about this - what range of heights do you mean. For example Fig 4b shows long-period oscillations to be absent (in NW winter/spring 2010 and 2012) only in a layer that is about 16-20km.

ll465-467: 'Since normally the BV frequency gets almost doubled near the tropical tropopause, it is almost impossible for long-period oscillations to penetrate through the tropopause in the tropical region.' - you should refer to some specific piece of theory to support this statement. (I have already expressed doubt that the Chen and Robinson work is directly relevant.)

ll478-494: This seems simply to be further general background material, adding to the background material already presented earlier in the paper. I don't see how it is directly relevant to the results in the paper.

l561: 'importance of the subtropical westerly jet' - be clear about which subtropical jet you mean, in particular at what level. Your discussion of Fig 11 seems to imply that you do not view the subtropical jet at 70hPa as particularly important. But the only other level you consider is 10hPa - and then is it really your intention to argue that this is exerting control over the entire tropical stratosphere above ∼20km (since you characterise only the 16-20km layer as where the long-period oscillations are absent from the tropical stratosphere).

l849, l852: 'Zieme' should be 'Ziemke'

---

## Referee Comment (RC2) · Anonymous Referee #2 · 15 Jun 2016

This manuscript is a follow-up paper of Guharay et al. [2004] cited in their reference list. Guharay et al. studied the intraseasonal variability (broadly defined as periods between 11 – 80 days) in zonal wind over Gadanki, India (13° N) using radiosonde, reanalysis and satellite data between surface and 100 km. Guharay et al. noted a drop of intraseasonal signal in the lower stratosphere and the signal reappears in the upper stratosphere and above. Guharay et al. did not provide any explanation why the intraseasonal variability exhibits such vertical structure over Guharay, but simply speculated a few possible mechanisms, including the Ziemke-Stanford mechanism, in which tropospheric intraseasonal variability first propagate poleward tropics near the

tropopause, then refracted back to tropical stratosphere. The current manuscript reexamines the same vertical structure using satellite temperature data. The authors show that, by further categorizing the intraseasonal variability into short (10 – 40 days) and long regimes (40 – 80 days), the long intraseasonal variability reappears in the upper stratosphere but not the short one. Simply assuming that Guharay et al.'s speculation about the Ziemke-Stanford mechanism is true, the authors further propose that the meriodional movement of the stratospheric subtropical jet is responsible for the equatorward propagation of the extratropical intraseasonal signals.

Unless I am missing something, I feel strongly that this paper is not well written and I cannot recommend it to be published in Atmos. Chem. Phys in the current form.

1. The introductory section is not well written. At least a thorough review of Guharay et al. [2004] should be provided. Otherwise, the reader has very hard time to figure out why the authors only look at India although they have access to global reanalysis data. Indeed, some writings in the current manuscript seem to be directly copied from the Introduction and Conclusion sections of Guharay et al. [2004].

2. The Ziemke-Stanford mechanism is only one of a few possible mechanisms mentioned in Guharay et al. [2004], which Guharay et al. did not provide any proof of its applicability to the zonal wind data. The current manuscript seems to have built solely on the assumption that the Ziemke-Stanford mechanism is correct.

3. The connection between the stratospheric subtropical jet and equatorward refraction is meant to support the validity of the Ziemke-Stanford mechanism for temperature, but such connection is not a strong evidence. The authors should at least analyze the E-P flux as in Ziemke and Stanford [1991].

4. There are too many grammatical mistakes.

---

## Author Comment (AC1) · 11 Aug 2016

Responses to the interactive comments By Prof. Haynes (Referee as well as Editor, Atmos. Chem. Phys. Discuss.,doi:10.5194/acp-2016-118-RC1, 2016) on

"Observational evidences of the influences of tropospheric subtropical and midlatitude stratospheric westerly jets on the equatorial stratospheric intraseasonal oscillations"

by G. Karthick Kumar Reddy et al.

**General Responses:**

We are indebted to the in depth comments of the reviewer, which indicates the large expectation of him/her from our present work. The below listed their four main reasons that led the reviewer to rejecting our manuscript can be handled by us effectively so that it can lead to get accepted for publication in the Atmospheric Chemistry and Dynamics of EGU publications. We provide below our one to one responses to the reviewer's comments and hope that the reviewer will give us a chance to revise our manuscript to the acceptable level of publication.

**Point by point responses:**

**Four main reasons**

(**1**)The new ingredient (relative to previous publications) is interannual variability, but only 4 years are considered. Therefore any identification of behaviour is highly speculative. (There is no modelling to support any of the ideas presented.)

**Response 1: We will add few more years of data to address this issue**

(**2**) It is difficult to see the value of the radio occultation data in this study. There is no argument that the phenomenon being studied has small vertical scales and therefore can be studied much more effectively with radio occultation data than with reanalysis data. Furthermore the radio occultation data is used only in a limited geographical region. Why has it not been used across the whole tropics and subtropics? (It seems as though the region of study has been used for historical reasons, this is for example that has been observed by an MST radar, but there is no use of radar data here.)

**Response 2: Radio occultation data of temperature are highly accurate to ~0.25°C in the heights of 8-20 km which is about twice the accuracy of normal radiosonde data. It is to be noted that the accuracy of the reanalysis data improves significantly with the inclusion of assimilation of data from RO technique. Further the height resolution of reanalyses data is much poorer when compared to the RO data. Why we should not get interested to find the characteristics of intraseasonal oscillations over a particular location. In such findings why it is stressed that we should also look into global scale characteristics of these oscillations. MST radar gives only wind velocity not temperature. To strengthen our point of view that analyses of ISO at particular**

locations will given in depth view of them, Figs. 1 and 2 below show the combined empirical orthogonal function (EOF) and wavelet analyses of the ERA-interim data of temperature at longitudes centered around 79°E (Gadanki, India) and 180°E for the whole Northern Hemisphere during 2009-2012. Figs. 1a and b (right panel) show the wavelet spectra of first seven eigen projections (principal components) of the time and latitude mappings respectively of the atmospheric temperature at 17 km. From top to bottom, the panels correspond to decreasing eigen values. It may be observed that the first principal component of time series (top left panel of Fig. 1) over the Indian region shows strong ~64 day oscillation in all the winters except for 2011. However, this is not the case for the Central Pacific region (Fig.2). Along with that the first principal component in latitude also shows strong ~64 day oscillation for almost all the latitudes (top right panel of Fig.2). However, there is no such ~64 day oscillation in latitude mapping over the Indian region (top right panel of Fig.1) indicating large variations between ISO activities in different longitude sectors.

**Fig.1**

[Figure]

**Fig. 2**

[Figure]

**(3)** Given the absence of any arguments that 'new' data such as radio occultation data is essential, there would apparently be no reason not to use ERA-interim data (for example) over the whole 30-year time period for which it is available - and that would be a much more satisfactory approach to studying inter annual variability. It would also allow various types of composite/correlation analysis of the type used by the two papers mentioned above.

**Response 3: The present work is an indepth analysis of intraseasonal oscillations in temperature at a particular location over the tropics. Presenting 30 years of data analysis in a single paper is difficult along with the present results.**

**(4)** The paper is not well written. I have given many comments below on this. I note in particular the large number of abbreviated terms defined. If the authors were to insist on retaining these then perhaps there should be a table summarising the definitions. Additionally there are a very large number of figures and these are ineffective in identifying and explaining the important points that the authors wish to make. (The number of the figures and their nature is simply unhelpful to the reader.) Therefore my recommendation as a referee is that this paper should be rejected for publication in ACP.

**Response 4: If it is given a chance, we will surely revise our manuscript to the level of acceptance in ACP.**

**Detailed comments:**

**(1)**MStWJ' - is this intended to be distinct from the stratospheric polar night jet?

**Yes.**

**(2)** 'December-May (Northern winter to summer, NWTS)' - this is one of many abbreviated terms that are introduced - my own view is that there are too many. In any case they need to be chosen to be as intuitive as possible - Northern winter and spring - NWS - might be better. But I suggest you simply say 'December-May' when needed.

**'December-May' is agreed.**

**(3)** there is significant' > 'there are significant'. **Agreed**

**(4)** The 40-80' > 'A 40-80' (because a 40-80 oscillation has not previously been introduced, nor, I believe, is it a standard term). **'A' is agreed**

**(5)** refracted' > refraction' **Agreed**

**(6)** the two longer period bands' - be explicit - 'the 20-40 and 40-80 period bands'
     **Out of three period bands given, "the two longer period bands" gives the explicit meaning.**

**(7)** l30-32: 'It is also observed that the phase of the : : : QBO : : : has significant control on the strength of the : : : MStWJ : : : that in turn controls the refraction' - I don't see how you can argue on the basis of the 4 years of observations presented (and nothing else) that the QBO has significant control on the stratospheric jet, nor do I see how you can argue that the stratospheric jet has control over refraction.

**Agreeing with the reviewer's comment, 'observed' is replaced with 'suspected.'**

**(8)** l33-34: 'LISO' and 'SISO' - two new abbreviations which in my opinion do not help the reader. **These are not my abbreviations but provided by the earlier publications which are referenced.**

**(9)** l54: 'island' > 'Island' **Agreed**

**(10)** Intra Seasonal Oscillations' - say explicitly what you mean by this term - I think any oscillations with period 10-80 days. **It is now 10-80 day intra seasonal oscillation.**

**(11)** l53-77: There is a lot of detail here about the mesosphere, which is not the main subject of the paper. Unless you can identify specific aspects of the mesosphere which relate closely to the method or conclusions of the paper then this material should be significantly shortened - as it stands it distracts the reader from what turns out to be the main focus of the paper. The same comment applies to l79-89 - though some of that relates to a possible QBO effect and is therefore a little more relevant.

**Agreeing with the reviewer, we will cut short most of them.**

**(12)** l64: 'It is reported' > 'It was reported' **Agreed**

**(13)** l82: 'that dissipating' > 'that the dissipating' **Agreed**

**(14)** You leap from 'MJO' in l95 to 'intraseasonal activity' in l97. Is that intended to make a distinction - e.g. is 'intraseasonal activity' broader then 'MJO'? **It is now MJO.**

**(15)** l108-110: 'response of 100-hPa level water vapour : : : is out of phase with that at 215 and 147hPa levels' - 'convectively active phase moistens the upper troposphere, the tropopause region becomes dryer' - aren't those two things the same?
**(16)** l112-115: It wasn't clear to me how this information on the water vapour signal in the lower stratosphere was relevant to what you are considering (which is more to do with the dynamics). All in all, this paragraph gives the sense of a slightly random set of facts about the MJO. More focus would help the reader.

**Sentences associated with the comments 15 and 16 are now removed.**

**(17)** l118-113: Again it wasn't clear to me that this material on high-latitude intraseasonal oscillations was relevant. Are you implying a physical connection/relevance to the low-latitude phenomenon - or is this simply describing another 'intraseasonal' (=lowfrequency) oscillation?

**Implying a physical connection/relevance to the low-latitude phenomenon.**

**(18)** l140: You refer here to the previous work of Ziemke and Stanford (1991) who considered an analogous problem in the southern hemisphere and Niranjan Kumar et al (2011) who considered vertical propagation of intraseasonal oscillations in the northern hemisphere. What I am missing is what new ingredients you are providing over the Niranjan Kumar et al (2011) paper.

**The new material is that with the inclusion of equatorial quasi-biennial oscillation phases, the present manuscript explains the interannual variation of the intraseasonal oscillations**

**(19)** l143: I think that by 'the subtropical westerly jet' here you mean a jet in the stratosphere - please be explicit. **Subtropical westerly jet is a well known tropospheric phenomenon**

**(20)** l149: 'whole tropical-high latitude regions' - clearer to say 'in the whole Northern Hemisphere'? **'Whole Northern Hemisphere' is agreed.**

**(21)** l152-168: This paragraph seems to a justification of presenting the information local to Gadanki as part of a study of what is a much larger scale phenomenon. You mention the QBO as an analogous example, but the analogy seems weak - the QBO is a largescale phenomenon that requires small-scale processes for its existence. I don't see any argument in your paper as written that the large-scale phenomenon of intraseasonal oscillations requires small-scale processes (that may be observed local to Gadanki) for its existence.

**New arguments in this way as you suggested will be provided**

**(22)** l197: 'wet temperature, wetPrf' - this is important technical detail but most readers will find it mystifying. Provide a reference that gives more information on these terms.

**Required reference will be provided**

**(23)** l199: GPS radio occultation potentially provides temperature observations over much larger regions than that you have chosen. Why did you not use all the available observations?

**Presenting many observations over larger regions in a single paper is difficult.**

**(24)** l265: Chen and Robinson (1992) considered the combined effects of vertical shear and change in buoyancy frequency at the extratropical tropopause. So I don't see that their paper can be used straightforwardly to explain what may or may not be happening at the tropical tropopause.

**Effects of vertical shear and change in buoyancy frequency are latitude independent.**

**(25)** l268: You should mention the black lines in the Figures in the captions as well as the text. If one set of black lines is supposed to indicate April and the other December then you should differentiate between the two - e.g. by making one set dashed.

**Now the Figures are clear**

**(26)** l292: 'near 120 days in 2009' - but this feature lies outside your bounding curves (at left=hand and right-hand ends of the figure) for the validity of the wavelet transform calculation. So how can you consider this feature to be important?
**Not all the part of this oscillation is outside the bounding curves**

**(27)** 295: 'this oscillation is present' - which oscillation exactly - _50 days or _120 days? You say 'at all other heights'. But at several heights, e.g. 38km and 40km, there are no crimson contours, indicating significance in all non-2011 years. (It would be very

helpful to have years marked on the Figures and I do not understand why this has not been done.)

**Now the figure and the descriptions are clear**

**(28)** l302: 'distinctly enhanced 40-80 day band oscillation : : : during the months of November-May in all the years' - only in 2 years surely. As you say yourself the signal is weak in 2011 (and, if 2009 is to be considered to be within the range of the data, there is very little signal in 2009).

**This sentence is slightly modified to give clearer meaning**

**(29)** l328: 'zonal wind shows clear poleward propagation (tilting towards the right side with increasing latitude)' - this tilt - which I suppose should be visible in the bottom-left panel of Figure 8 - seems very weak to me. You note that the wind fluctuations (apparently) show phase tilt, but the temperature does not. How do you rationalise that with a dynamical mechanism? Are the temperature and wind signals independent - i.e. showing essentially independent dynamical phenomena? Are both signals 'real'?

**This is one of the main reasons why particularly the temperature data from RO satellites are taken for the present study. We will stress more on this aspect in the revised manuscript.**

**(30)** l343: You have previously described a phenomenon of refraction back towards the tropics above 19km (see ll317-321). But here you are using the wind at 30km as an explanation for absence of refraction in 2011. Why should the wind at 30km affect what is happening at 19km upwards?

**We now explain more in this regard to give clearer meaning for different period bands of intraseasonal oscillations**

**(31)** l346: It is not clear to me that invoking the Arctic Oscillation (in discussing the wind variations at 10hPa) here is very helpful. The Arctic Oscillation is primarily a tropospheric phenomenon in which the mid-latitude eddy-driven jet shifts (with implications for, e.g. sea-level pressure at high latitudes). It is true that stratospheric variations are discussed in terms of an NAM (Northern Annular Mode), but that is simply the dominant pattern of dynamical variability, in the stratosphere or at some other chosen level. In the troposphere the NAM is associated with the Arctic Oscillation, but that does not imply that the association can be made at all levels.

**We will include more discussions with proper references to make the things more clear.**

**(32)** l348: You seem to be considering the zonal velocity at 70hPa simply to show that it does not show anomalous behaviour in 2011 relative to other years, in contrast to the zonal velocity at 10hPa. **Yes**

**(33)** l354: 'no significant LISO in : : : 2011 in the heights of 30-40km' - if the intention is to focus on what is happening above 30km then that should have been made clear at the beginning of this paragraph. But the fact is that you have previously characterised the absence of LISO by referring to Fig 4b - in which inter annual differences are shown in the layer 19-32km.

**This paragraph is now elaborated such that clearer meaning can be obtained.**

**(34)** l358: '50-day oscillation' - are you using this term to be intentionally different from previous use of '40-80-day'? **No**

**(35)** l383: You have focused here on the phase behaviour of the 40-80 day fluctuations. But one thing that is striking about Figs 12-15 is that the amplitude (e.g. in Jan-May) shows no sign of the claimed disappearance of the oscillation in the tropics in a region just above the tropopause. So how is the amplitude behaviour you show consistent with your previous discussion and indeed with the theme of the paper as a whole.

**Agreeing with the reviewer, it is now discussed more here to provide convincing results.**

**(36)** l413: 'increasing trend' is a very confusing term to use to describe increase with height. **Appropriate phrase is now used**

**(37)** l460: 'in the first few kilometers of height' - be more precise about this - what range of heights do you mean. For example Fig 4b shows long-period oscillations to be absent (in NW winter/spring 2010 and 2012) only in a layer that is about 16-20km.

**Now it is written more clear**

**(38)** ll465-467: 'Since normally the BV frequency gets almost doubled near the tropical tropopause, it is almost impossible for long-period oscillations to penetrate through the tropopause in the tropical region.' - you should refer to some specific piece of theory to support this statement. (I have already expressed doubt that the Chen and Robinson work is directly relevant.)
**Effects of vertical shear and change in buoyancy frequency are latitude independent.**

**(39)** ll478-494: This seems simply to be further general background material, adding to the background material already presented earlier in the paper. I don't see how it is directly relevant to the results in the paper

**These sentences are now removed.**

**(40)** l561: 'importance of the subtropical westerly jet' - be clear about which subtropical jet you mean, in particular at what level. Your discussion of Fig 11 seems to imply that you do not view the subtropical jet at 70hPa as particularly important. But the only other level you consider is 10hPa - and then is it really your intention to argue that this is exerting control over the entire tropical stratosphere above _20km (since you characterise only the 16-20km layer as where the long-period oscillations are absent from the tropical stratosphere).

**This paragraph is rewritten such that clearer meaning can be now obtained**

**(41)** l849, l852: 'Zieme' should be 'Ziemke' **Agreed**

---

## Author Comment (AC2) · 11 Aug 2016

Responses to the interactive comments Referee#2 (Atmos. Chem. Phys. Discuss., doi:10.5194/acp-2016-118-RC2, 2016) on

"Observational evidences of the influences of tropospheric subtropical and midlatitude stratospheric westerly jets on the equatorial stratospheric intraseasonal oscillations"

by G. Karthick Kumar Reddy et al.

**General Responses:**

We express our sincere thanks to the reviewer for the critical comments. We provide below our one to one responses to the reviewer's comments and hope that the reviewer will give us a chance to revise our manuscript to the acceptable level of publication in ACP journal.

**Point by point responses:**

This manuscript is a follow-up paper of Guharay et al. [2004] cited in their reference list. Guharay et al. studied the intraseasonal variability (broadly defined as periods between 11 – 80 days) in zonal wind over Gadanki, India (13_ N) using radiosonde, reanalysis and satellite data between surface and 100 km. Guharay et al. noted a drop of intraseasonal signal in the lower stratosphere and the signal reappears in the upper stratosphere and above. Guharay et al. did not provide any explanation why the intraseasonal variability exhibits such vertical structure over Guharay, but simply speculated a few possible mechanisms, including the Ziemke-Stanford mechanism, in which tropospheric intraseasonal variability first propagate poleward tropics near thetropopause, then refracted back to tropical stratosphere. The current manuscript reexamines the same vertical structure using satellite temperature data. The authors show that, by further categorizing the intraseasonal variability into short (10 – 40 days) and long regimes (40 – 80 days), the long intraseasonal variability reappears in the upper stratosphere but not the short one. Simply assuming that Guharay et al.'s speculation about the Ziemke-Stanford mechanism is true, the authors further propose that the meriodional movement of the stratospheric subtropical jet is responsible for the equatorward propagation of the extratropical intraseasonal signals.

Unless I am missing something, I feel strongly that this paper is not well written and I cannot recommend it to be published in Atmos. Chem. Phys in the current form.

**Response 1: If chance is given we will attempt to revise the manuscript to the acceptable level of publication in ACP journal.**

(**2**) 1. The introductory section is not well written. At least a thorough review of Guharay et al. [2004] should be provided. Otherwise, the reader has very hard time to figure out why the authors only look at India although they have access to global reanalysis data. Indeed, some writings in the current manuscript seem to be directly copied from the Introduction and Conclusion sections of Guharay et al. [2004].

**Response 2: Global reanalyses data do not have such high resolution in vertical as the RO data have. Hence it is not possible to find exactly at what height near the tropopause the ISO signals are refracting towards subtropical latitudes.**

**(3)** 2. The Ziemke-Stanford mechanism is only one of a few possible mechanisms mentioned in Guharay et al. [2004], which Guharay et al. did not provide any proof of its applicability to the zonal wind data. The current manuscript seems to have built solely on the assumption that the Ziemke-Stanford mechanism is correct.

**Response 3: The present work is an extension of previous works to test whether the Ziemke-Stanford mechanism works all times and other longitudinal (India) also.**

**(4)** 3. The connection between the stratospheric subtropical jet and equatorward refraction is meant to support the validity of the Ziemke-Stanford mechanism for temperature, but such connection is not a strong evidence. The authors should at least analyze the E-P flux as in Ziemke and Stanford [1991].

**Response 4: If it is given a chance, we will surely revise our manuscript uitilizing E-P flux**

(**5**) 4. There are too many grammatical mistakes.

Response 5: Grammatical mistakes can be easily removed.